# Arabidopsis ASYMMETRIC LEAVES2 and Nucleolar Factors Are Coordinately Involved in the Perinucleolar Patterning of AS2 Bodies and Leaf Development

**DOI:** 10.3390/plants12203621

**Published:** 2023-10-19

**Authors:** Sayuri Ando, Mika Nomoto, Hidekazu Iwakawa, Simon Vial-Pradel, Lilan Luo, Michiko Sasabe, Iwai Ohbayashi, Kotaro T. Yamamoto, Yasuomi Tada, Munetaka Sugiyama, Yasunori Machida, Shoko Kojima, Chiyoko Machida

**Affiliations:** 1Graduate School of Bioscience and Biotechnology, Chubu University, Kasugai 487-8501, Japan; sando@isc.chubu.ac.jp (S.A.); iwakawa@se.kanazawa-u.ac.jp (H.I.); gr15803-5749@sti.chubu.ac.jp (S.V.-P.); vf.19s.3654@a.thers.ac.jp (Y.M.); 2Division of Biological Science, Graduate School of Science, Nagoya University, Nagoya 464-8602, Japan; nomoto@gene.nagoya-u.ac.jp (M.N.); luolilan@genetics.ac.cn (L.L.); ytada@gene.nagoya-u.ac.jp (Y.T.); 3Center for Gene Research, Nagoya University, Nagoya 464-8602, Japan; 4Department of Biology, Faculty of Agriculture and Life Science, Hirosaki University, Bunkyo-cho, Hirosaki 036-8561, Japan; msasabe@hirosaki-u.ac.jp; 5Department of Life Sciences, National Cheng Kung University, Tainan City 701, Taiwan; 10902017@gs.ncku.edu.tw; 6Division of Biological Sciences, Faculty of Science, Hokkaido University, Sapporo 060-0810, Japan; 7Department of Biological Sciences, Graduate School of Science, The University of Tokyo, Tokyo 113-0033, Japan; sugiyama@bs.s.u-tokyo.ac.jp

**Keywords:** 45S ribosomal DNA, AS2/LOB domain, AS2/LOB family, DEAD-box helicase, nucleolin, zinc finger, nucleolus

## Abstract

Arabidopsis ASYMMETRIC LEAVES2 (AS2) plays a key role in the formation of flat symmetric leaves. AS2 represses the expression of the abaxial gene *ETTIN*/*AUXIN RESPONSE FACTOR3* (*ETT*/*ARF3*). AS2 interacts in vitro with the CGCCGC sequence in *ETT*/*ARF3* exon 1. In cells of leaf primordia, AS2 localizes at peripheral regions of the nucleolus as two AS2 bodies, which are partially overlapped with chromocenters that contain condensed 45S ribosomal DNA repeats. AS2 contains the AS2/LOB domain, which consists of three sequences conserved in the AS2/LOB family: the zinc finger (ZF) motif, the ICG sequence including the conserved glycine residue, and the LZL motif. *AS2* and the genes *NUCLEOLIN1* (*NUC1*), *RNA HELICASE10* (*RH10*), and *ROOT INITIATION DEFECTIVE2* (*RID2*) that encode nucleolar proteins coordinately act as repressors against the expression of *ETT*/*ARF3.* Here, we examined the formation and patterning of AS2 bodies made from *as2* mutants with amino acid substitutions in the ZF motif and the ICG sequence in cells of cotyledons and leaf primordia. Our results showed that the amino acid residues next to the cysteine residues in the ZF motif were essential for both the formation of AS2 bodies and the interaction with *ETT*/*ARF3* DNA. The conserved glycine residue in the ICG sequence was required for the formation of AS2 bodies, but not for the DNA interaction. We also examined the effects of *nuc1*, *rh10*, and *rid2* mutations, which alter the metabolism of rRNA intermediates and the morphology of the nucleolus, and showed that more than two AS2 bodies were observed in the nucleolus and at its periphery. These results suggested that the patterning of AS2 bodies is tightly linked to the morphology and functions of the nucleolus and the development of flat symmetric leaves in plants.

## 1. Introduction

ASYMMETRIC LEAVES2 (AS2) is a member of the AS2/LOB family, which has a plant-specific AS2/LOB domain [1,2,3,4]. The AS2/LOB domain is widely conserved in plants ranging from algae to land plants, but is not found in animal and fungi kingdoms [5,6]. In *Arabidopsis thaliana*, 42 predicted proteins are classified in the AS2/LOB family [1,4]. The AS2/LOB domain consists of three sequences conserved in all members of the family, that is, the zinc finger (ZF) motif; the internal sequences containing conserved glycine residue (ICG sequence), and the leucine zipper-like (LZL) motif. The AS2 gene is the first cloned gene among the members of the AS2/LOB family [1], and has been most intensively investigated at both the genetic and molecular levels [3,7,8]. Elucidating the molecular functions of the three conserved sequences of AS2 will contribute to the understanding of the family members.

Leaves develop from a shoot apical meristem (SAM) as lateral organs along the three axes: proximal–distal, adaxial–abaxial, and medial–lateral [9,10,11,12,13]. The direct target of AS2 has been clarified via ChIP analysis and transcriptome analysis [14,15,16]. AS2 forms a complex with the myb protein AS1, and is involved in the epigenetic repression of the abaxial genes *ETTIN*/*AUXIN RESPONSE FACTOR3* (*ETT*/*ARF3*), *ARF4*, and class 1 *KNOX* homeobox genes, resulting in the establishment of the three morphological axes of leaves [14,17,18,19]. AS2 regulates in the repressive manner of *ETT*/*ARF3* expression in multiple pathways; that is, in one pathway it directly binds to the promoter region of the *ETT*/*ARF3*, and in the other it indirectly represses the gene expression by activating the tasiR-ARF pathway [14,16]. In addition, AS2-AS1 is involved in maintaining DNA methylation on the *ETT*/*ARF3* gene [14,20]. Furthermore, it has been shown that AS2 interacts with sequences containing GCGGCG/CGCCGC in *ETT*/*ARF3* exon 1 [20,21]. The ZF motif of the AS2/LOB domain is required for the AS2 protein–*ETT* DNA interaction, via the above sequences in the *ETT*/*ARF3* gene [20].

The AS2-AS1 complex directly represses the transcription of the class 1 *KNOX* genes (*BP*, *KNAT2*, *KNAT6*), resulting in proximal–distal axis elongation [18,19]. AS2-AS1 also mediates the stable repression of the class 1 *KNOX* genes in developing leaves via the recruitment of Polycomb Repressive Complex2 (PRC2), which methylates lysine 27 in histone H3 [16,22]. On the other hand, the histone H3K27me3 modification of the *ETT*/*ARF3* genome region has not been reported. Although AS2-AS1 is required for the correct temporal repression of *ETT*/*ARF3*, its mechanism might be independent from the PRC2 pathway [16]. Other different mechanisms have also been suggested for the maintenance of *ETT* repression, but the details of such mechanisms remain to be investigated.

In cells of the adaxial layer of leaves of Arabidopsis, the AS2 protein is localized in the nucleoplasm and peripheral regions of the nucleolus and forms bodies, named AS2 bodies [23,24,25]. The AS2 bodies are partially overlapped with chromocenters including those of condensed 45S rDNA repeats at the perinucleolar region [25].

The nucleolus is the most distinctive nuclear sub-compartment, and is the transcription site of ribosomal RNA (rRNA) gene, their processing, and ribosome biogenesis [26]. In addition, it also plays a variety of roles in cell cycle control and stress responses. Recently, in human cells, it has been reported that heterochromatin interactions organize around the nucleolus, acting as an inactive interchromosomal hub [27,28]. The nuclear periphery and nucleolus are nuclear landmarks that contribute to heterochromatin signiures. Although the role of the nuclear membrane (nuclear lamina) in chromatin organization is well known, the function of the nucleolus remains unknown.

Various mutations (about 60) that markedly enhance defects in the development of the adaxial domain of leaves in the *as2* or *as1* background have been reported [7,8]. The genes responsible for these mutations are referred to as “modifiers” or modifier genes, which affect the expression of genes controlled by *AS2* and *AS1* [7]. Double mutants generate abaxialized filamentous leaves that have lost the adaxial domain. In many cases, causative mutations occur in genes encoding nucleolar proteins, such as *NUCLEOLIN1* (*NUC1*), *RNA HELICASE10* (*RH10*), and *ROOT INITIATION DEFECTIVE2* (*RID2*), as well as genes encoding proteins involved in chromatin modification, the biogenesis of small RNAs, DNA replication, and ribosomal proteins [7,8,29,30,31,32,33,34,35,36,37]. The Arabidopsis *RH10* gene, which potentially is an ortholog of yeast Rrp3 and human DDX47, belongs to the DEAD-box RNA helicase family [31]. This helicase family has an indispensable role in gene regulation through RNA metabolism, and many members of this family locate in the nucleolus. It has been reported that altered rRNA biogenesis and nucleolar structures are altered in *nuc1*, *rh10,* and *rid2* mutants of Arabidopsis [31,38,39]. However, the relationship between the functions of these genes and the formation of AS2 bodies is yet unknown.

We have shown that the ZF motif of AS2, which is required for binding to the coding DNA of *ETT*/*ARF3* [20], is essential for the formation of AS2 bodies in adaxial epidermal cells, but not for nuclear localization. Furthermore, the regions encompassing the ICG sequence and the LZL motif are essential for the nuclear localization of AS2 [25]. Although we reported that the AS2/LOB domain is responsible for the formation and functions of AS2 bodies, the roles of amino acid residues conserved in the AS2/LOB family have been unknown in relation to the formation of AS2 bodies at perinucleolar regions and the binding to *ETT*-DNA.

In the present study, we found two new *as2* mutant alleles showing the typical leaf phenotype of *as2*: they have amino acid replacements in their ZF motif or conserved glycine residues, respectively. We examined the perinucleolar localization of the mutant as2–YFP fusion and molecular interactions with the target sequence in *ETT.* The results showed that the amino acid residues neighboring four conserved cysteine residues were required for both the formation of perinucleolar bodies and for interactions with the CGCCGC sequence in *ETT* exon 1. The conserved glycine residue was also required for the formation of perinucleolar bodies, but not for the interaction with the CGCCGC sequence. In addition, we examined the effects of *nuc1-1*, *rh10-1*, and *rid2-1* mutations of Arabidopsis on the formation of AS2 bodies. The results showed that these mutations altered the numbers, sizes, and patterning of AS2 bodies in the nucleolus. These observations suggest that the plant nucleolus provides critical and major constituents to form AS2 bodies that play an essential and functional role in leaf development.

## 2. Results

### 2.1. Characterization of Mutants Containing Amino Acid Replacement in the ZF Motif or the ICG Sequence of the AS2/LOB Domain

To know which amino acid residues of the AS2/LOB domain are responsible for the perinucleolar localization of AS2 bodies and the binding with *ETT*-*DNA*, we used *as2* mutants with amino acid substitutions in the ZF motif or the conserved glycine residue in the ICG sequence.

The *snp1* mutant was originally isolated as a mutant that suppressed the leaf phenotype of the *nph4* mutant, and was named the “suppressor of *nph4*” [40]. The *snp1* single mutant showed downwardly curled leaves and leaf lobes that resembled the shape of leaves in the *as2-1* mutant. A genetic analysis showed that *snp1* was a recessive single mutation. The mutant had two base changes which caused two amino acid substitutions, P9S and A11V, in AS2 (Figure 1A). The P9S and A11V were adjacent to the first cysteine residue, C10, among the four cysteine residues in the zinc finger motif of the AS2/LOB domain (Figure 1A). The mutations in AS2 affect several leaf phenotypes, such as downwardly curled leaves and the formation of leaf lobes [41]. We examined the fifth and sixth leaves of Col-0, the *as2-1,* mutant and the *snp1* mutant (Appendix A). Similarly, as in *as2-1*, the fifth and sixth leaves of *snp1* showed asymmetric leaf lobes (Appendix A). We crossed *snp1* with *as2-1* and analyzed the leaf morphology in the F_1_ plants. The leaf lobes in F_1_ were found to be similar to those in *as2-1*. These results suggested that the abnormal leaf phenotype of *snp1* was caused by both or either one of the amino acid changes in AS2. Therefore, we renamed *snp1* to be *as2-snp1* as one of the alleles of *as2*. AS2 is involved in the repression of the expression of the abaxial determinant genes, class 1 *KNOX* genes, *IPT3*, and *KRP5*. We investigated the transcript levels of the previously mentioned genes via quantitative reverse transcriptional-PCR (qRT-PCR; Appendix A). The mRNA levels of abaxial genes such as *ETT*/*ARF3*, *FIL*, *YAB5*, *KAN2*, and class 1 *KNOX* genes, *IPT3*, and *KRP5,* increased in the *as2-snp1* mutant as compared with those in Col-0, as shown in *as2-1* in previous reports [14,15,19,41,42]. These results were consistent with our previous report, and suggested that the abnormal leaves in the *as2-snp1* mutant were abaxialized and defective in the sizes of the leaves, as shown in *as2-1* [14,19].

In the *as2-5* mutant, a single guanine nucleotide at position 137 was replaced with an adenine nucleotide in the coding region for the amino-terminal (N-terminal) half of the AS2/LOB domain, resulting in the substitution of the glycine residue at position 46 by a glutamic acid residue in the ICG sequence (Figure 1B) [1]. This glycine residue is essential for the function of AS2. Since G46 in AS2 is conserved in the corresponding position of all members of the AS2/LOB family [1,2,3,4,43], the 46th glycine residue should be essential for functions of the AS2/LOB family proteins. As described previously [41] and shown in Appendix A, *as2-5* generated leaf lobes, and the genetic background of *as2-5* is L*er*-0.

### 2.2. Amino Acid Residues in the ZF Motif and the Conserved Glycine Residue in the ICG Sequence Are Required for the Formation of AS2 Bodies

We have shown that amino acid residues conserved in the ZF motif are critical for the formation of AS2 bodies [25]. In cells of the adaxial domain of leaves and cotyledons, AS2 fused with yellow fluorescent protein (YFP) is localized to the nucleoplasm and two nuclear bodies, named AS2 bodies, which are localized in the inner margin of the nucleolus (Figure 2A) [23,25]. The AS2/LOB domain is sufficient for the formation of AS2 bodies, and the four cysteine residues in the ZF motif are required for forming AS2 bodies [25].

We constructed transgenes that encode YFP fused to the C-terminals of full-length AS2 or as2-snp1 and introduced these constructs into Col-0 plants. These proteins were expressed in the transformants, and the signals due to YFP were observed in cells at the adaxial surfaces of cotyledons and leaves. The nuclei were visualized through DAPI staining. The nucleoplasm and the nucleolus were distinguished based on the DAPI-positive and DAPI-negative (dark) areas, respectively [23,25]. As shown in Figure 2B, the signals due to as2-snp1-YFP (P9S_A11V) were detected in the nucleoplasm, but signals due to YFP were not detected as subnuclear bodies. These results suggest that the proline and/or alanine residues adjacent to the cysteine residue are essential for the formation of AS2 bodies.

To investigate which amino acid residues in the ZF motif in the AS2/LOB domain are crucial for the formation of AS2 bodies, we performed domain-swapping experiments. The amino acid sequences of the ZF motif of the AS2/LOB domain are well conserved among AS2 family members, but the three amino acid sequences between C20 and C24 (the third interval) are diverse [1,3]. AS2 encodes the QPE sequence in the third interval, and AS2-like protein3 (ASL3)/LOB domain protein25 (LBD25), which is one of the closest members of the AS2/LOB family, encodes the TSD sequence [1,2,4]. We have reported that the ZF motif of ASL3/LBD25 cannot fully replace the functions of the AS2/LOB domain of AS2 in leaf development. The length of *as2* leaves in the proximal–distal direction was restored to the wild-type values in *as2-1*/as2(Q21T_P22S_E23D) [2]. However, the asymmetric formation of deep leaf lobes and leaflet-like structures were not recovered [2]. As shown in Figure 2C, no AS2 body was detected in 73% of the nuclei of the as2(Q21T_P22S_E23D)-YFP transgenic lines. Although the AS2 body-like signals were detected in 27% of the nuclei of as2(Q21T_P22S_E23D)-YFP, the size of the bodies was small and the YFP signals were very weak. In some cases, the AS2 bodies were not at the periphery, but inside of the nucleolus.

In the AS2 of monocotyledonous plants such as maize and rice, the QPE in the third interval is the QPD [44]. Therefore, we investigated whether the AS2 body is formed when E23 in *A. thaliana* is changed to aspartic acid. The AS2 bodies were observed in 87% of the nuclei of as2(E23D)-YFP transgenic lines at the periphery of the nucleolus, similar to AS2-YFP (Figure 2D). Q21 and P22 in the third interval may be important for AS2 body formation, and it may be sufficient if the 23rd amino acid is an acidic amino acid.

We next examined the function of G46 in the ICG sequence, which is conserved in all members of the AS2/LOB family. As shown in Figure 2E, no AS2 body was detected in any nuclei of the as2-5-YFP transgenic lines, although a YFP signal was detected in the nucleoplasm. This result suggested that G46 was necessary for the formation of AS2 bodies.

### 2.3. The AS2-Specific Amino Acid Sequences in the ZF Motif Are Required for the Interaction with the CGCCGC Sequence in exon 1 of ETT/ARF3

Previously, we reported that the wild-type AS2D protein (amino acid residues from 1 to 119 of AS2 protein) which was tagged with FLAG at the C-terminus of the AS2/LOB domain (designated AS2D-FLAG) interacts with the sequence in exon 1 of *ETT*/*ARF3* containing CGCCGC at the position of +264 to 269 (Ex1_264) (Figure 3A,B) [20]. The four cysteine residues in the ZF motif are required for the protein–double stranded DNA (dsDNA) interaction. To examine whether the two amino acid residues, which are replaced in the *as2-snp1* mutant, are involved in the protein–dsDNA interaction, as2-snp1D-FLAG was constructed and synthesized *in vitro* using the wheat germ system. Then, the protein–dsDNA binding experiments were conducted using the AlphaScreen system (Figure 3C; Section 4.7).

We used AS2D-FLAG and wheat germ extract (WGE) as the positive and negative controls, respectively. Biotinylated/non-biotinylated and synthesized 50-nucleotide dsDNA coding *ETT* exon 1 was used as the dsDNA (Figure 3B; Section 4.7). In our experiments, when the tagged proteins bind to the biotinylated dsDNA, the fluorescent signals from the acceptor beads should be highly amplified. When non-biotinylated dsDNA are used, the signals might not be amplified. When the AS2D-FLAG protein and Ex1_264 dsDNA were co-incubated, a high relative signal value (biotinylated DNA/non-biotinylated DNA) was detected, which was consistent with our previous report (Figure 3A(a),C(a)) [20]. When WGE, as the negative control, was incubated with the dsDNA, we obtained a very low relative signal value (Figure 3C(g)). Also, when as2-snp1D-FLAG and the dsDNA were incubated, the relative signal value we observed was as low as that obtained from the negative control (Figure 3A(b),C(b)). This result suggests that the interaction between the AS2D protein and Ex1_264 dsDNA requires at least one of the two amino acid residues adjacent to the first cysteine residue, both of which were replaced in the *as2*-*snp1* mutant.

The incubation of the mutant protein as2(Q21T_P22S_E23D) and the dsDNA including Ex1_264 did not generate detectable signal levels (Figure 3A(c),C(c)), whereas a high signal level was detected with the incubation of the mutant protein as2(E23D)D-FLAG and the dsDNA (Figure 3A(d),C(d)). Q21 and P22 in the third interval may be important for interactions with the dsDNA including Ex1_264, and it may be sufficient if the 23rd amino acid residue is an acidic amino acid. These results correlated with the formation of AS2 bodies.

We next examined the interaction of as2-5D-FLAG and dsDNA including Ex1_264, and detected a high signal level (Figure 3A(f),C(f)). This result suggested that binding to dsDNA including Ex1_264 is not a sufficient condition for the formation of the AS2 body, because as2-5-YFP (G46D) did not form any AS2 body, as shown in Figure 2E.

We examined whether AS2D-FLAG could physically bind to the biotinylated Ex1_264 DNA via a pull-down assay with streptavidin-conjugated donor beads (see Section 4.8). The result showed the physical interaction with the Ex1_264 dsDNA for AS2D-FLAG, as2(E23D)D-FLAG, and as2-5D-FLAG (Figure 3D(a,d,f)). These results suggest that the AS2 domain binds to the Ex1_264 DNA, which is mediated by the interaction between the ZF motif of AS2 and the core motif CGCCGC. The interaction of as2-snp1D-FLAG to the Ex1_264 DNA was detected in the pull-down assay very weakly (Figure 3D(b)), although the incubation of as2-snp1D-FLAG and dsDNA including Ex1_264 did not generate detectable signal levels. The binding of as2-5D-FLAG to the Ex1_264 DNA was detected with the pull-down assay (Figure 3D(f)). These results are correlated with the results of the AlphaScreen assay, in that the interaction of as2-5D-FLAG and Ex1_264 was detected at a high level (Figure 3C).

### 2.4. NUC1, RH10, and RID2 Are Involved in Subnucleolar Patterning of AS2 Bodies

Figure 2E and Figure 3D(f) showed that as2-5-YFP did not form AS2 bodies, but an interaction with dsDNA including the Ex1_264 sequence was detected. These results suggested that the formation of AS2 bodies may need some factor(s) other than Ex1_264 DNA, such as proteins or nucleic acids interacting with AS2 or/and affecting the conformation of AS2 proteins. It has already been reported that AS1 forms a complex with AS2 [45,46], but is not required for the formation of AS2 bodies (Figure 4A(b)) [25].

We next examined the AS2 body formation in the modifier mutants, *nuc1-1*, *rh10-1,* and *rid2-1*. As reported previously, the expression of the *ETT*/*ARF3* gene is influenced by factors that are localized in the nucleolus [31]. The AS2-AS1 complex binds directly to the upstream region of the leaf abaxial gene *ETT*/*ARF3* to repress its transcription [14,15,16]. Such transcriptional repression is further reduced by mutations in various genes for nucleolus-localized proteins, such as NUC1, RH10, and RID2, which are involved in the biogenesis of ribosomal RNAs and the formation of nucleoli with a normal morphology [31,38,39]. As shown in Figure 4B, the subnucleolar patterns of AS2 bodies in *nuc1-1*, *rh10-1,* or *rid2-1* were very different from those in the wild type. The number of signals of AS2-YFP increased in the *nuc1-1* mutant at a peripheral region of the nucleolus, and some AS2 bodies did not co-localize with the perinucleolar chromocenters (Figure 4B(a) and Appendix A) [25]. In the *rh10-1* mutant or *rid2-1*, AS2-YFP signals were detected, with many clusters, which varied in size, at the nucleolus periphery and inside the nucleolus (Figure 4B(b,c) and Appendix A). These results suggested that NUC1, RH10, and RID2 are involved in the patterning and/or maintenance of AS2 bodies at two domains of the nucleolus periphery.

## 3. Discussion

### 3.1. A Manner of Interaction of the Amino Acid Sequence (QPE) in the ZF Motif of AS2 Appears to Be Different from That of the ETT/ARF3 Interaction with CGCCGC and the Formation of AS2 Bodies

Figure 5A summarizes the abilities of *as2* mutants to form AS2 bodies and interact with the CGCCGC sequence and their leaf phenotypes. The four cysteine residues of the ZF motif and the intervals between each pair of cysteine residues are conserved among all the AS2/LOB family members, and the amino acid residues other than the cysteine residues in the ZF motif are similar among the members, except for those in the third interval [1,4].

In *as2-snp1* and *as2(4C/4A)*, the abilities to form AS2 bodies and to interact with CGCCGC were completely abolished (Figure 2 and Figure 3) [25]. The *as2-snp1* plant exhibited phenotypes of the typical loss-of-function mutant, such as *as2-1* (Appendix A). The *as2(4C/4A)* mutant allele did not exhibit the ability to complement the loss-of-function mutant *as2-1.* Thus, these two mutant alleles lead to null phenotypes of the *AS2* gene.

Next, we investigated the function of the three amino acid residues in the third interval of AS2. ASL3/LBD25 encodes the TSD sequence in the third interval. In the as2(Q21T_P22S_E23D)-YFP transgenic line, AS2 bodies were detected in 27% of the nuclei, although its signals were weak (Figure 2C). On the other hand, the CGCCGC interaction ability was below the detection limit (Figure 3C(c)). This result suggests that binding to *ETT* exon 1 is not necessarily involved in the formation of the AS2 bodies. Binding to a specific DNA region (including CGCCGC) is thought to be necessary for AS2 body formation, but identifying such DNA regions is a future issue.

We have already reported that the introduction of a chimeric as2(Q21T_P22S_E23D) into *as2* mutants restores the establishment of the proximal–distal axis through the suppression of class 1 *KNOX* expression in the *as2* mutant phenotype, but does not restore the establishment of the adaxial–abaxial axis through the suppression of *ETT*/*ARF3* expression [2]. Taken together, these results suggest that the repression of class 1 *KNOX* by AS2 may not require AS2 body formation. AS2-AS1 mediates the stable repression of class 1 *KNOX* genes, *BP,* and *KNAT2* in developing leaves via the recruitment of Polycomb Repressive Complex2 (PRC2) [16,22]. The *ETT*/*ARF3* coding sequence and its upstream sequence reveal no H3K27me3 deposition, and it was also suggested that AS2-AS1 is involved in a distinct and PRC2-independent mechanism for the stable repression of *ETT*/*ARF3* [16]. Our present results, together with the previous data, suggest that AS2-AS1 is involved in the maintenance of repression of *ETT*/*ARF3* via the formation of the AS2 bodies through a novel mechanism (Figure 5C) [2,14].

We have reported that AS2 is involved in maintaining DNA methylation in the coding region of *ETT*/*ARF3* [14,20]. Regarding the relationship between the maintenance of DNA methylation and the formation of AS2 bodies, it might be necessary to clarify what kind of mechanism is involved in the maintenance of the repression of *ETT*/*ARF3*.

The three amino acid residues in the third interval of the ZF motif may determine the sequence specificity of the binding DNA. Among the 42 members of Arabidopsis, only four groups have the same amino acid sequences. ASL1/LBD36 and ASL2/LBD10 contain TQE, which has been reported to be a factor involved in pollen development [47,48]. ASL37/LBD40, ASL39/LBD37, and ASL40/LBD38 contain SEN amino acid sequences, which are factors reported to be involved in the expression of nitrate-responsive genes [49,50]. ASL13/LBD24 and ASL14/LBD23 contain PKD; ASL33/LBD5 and ASL34/LBD8 contain PRF. Each group might recognize the same DNA sequence and be functionally related. Use of the ASL nomenclature might provide an edge to the discussion of the evolutionary developmental biology (evo-devo) of the AS2/ LOB family [1,2,3].

### 3.2. NUC1, RH10, and RID2 Are Necessary to Form AS2 Bodies at Two Peripheral Regions of the Nucleolus

Our next question is what factor(s) are necessary for the formation of AS2 bodies. We have already reported that AS1, which forms a complex with AS2, colocalizes with AS2 bodies, but is not involved in AS2 body formation (Figure 4A(b)) [25]. AS1 interacts with AS2 [18,45], and the leaf phenotype of the *as2 as1* double mutant plants does not differ from that of the *as2* or *as1* single mutant. The pattern of the AS2 bodies in *as1-1* was the same as those in the wild-type.

Figure 5B showed a summary of the leaf phenotypes of the double mutants with *as2* and the distribution patterns of the AS2 bodies in the mutants. In this study, we investigated the AS2 body formation in *nuc1-1*, *rh10-1*, and *rid2-1* mutants, which are all modifiers of *as2* mutants. The AS2 bodies were detected in each of three mutants, but interestingly, their patterns of localization at the nucleolus were found to be different from that of the wild type (Figure 4B).

Double mutants of *as2* with such mutations as *nuc1*, *rh10*, and *rid2*, synergistically up-regulate the abaxial genes, which generate abaxialized filamentous leaves with the loss of the adaxial domain. (Figure 5B) [31]. NUC1, RH10, and RID2 proteins are localized in the nucleolus and participate in pre-rRNA processing [31,39,51].

Intermediates of rRNA processing have been shown to accumulate in the *nuc1-1*, *rh10-1*, and *rid2-1* mutants [31,39,52]. Furthermore, structural abnormalities in the nucleolus are detected. In the *nuc1-1* plants, the nucleolus is disorganized and a disrupted internal structure is detected [38]. Nucleoli are enlarged in both the *rh10-1* and *rid2-1* mutants [31,39,51]. In these three mutants, AS2-YFP signals were detected as many clusters, which varied in size at the nucleolus periphery and inside the nucleolus (Figure 4B(a–c)).

FISH analysis by using 45S rDNA sequences as a probe showed that rDNA repeats seem to be decondensed in all nuclei in the *nuc1-1* mutant when compared with those in the wild type [38]. In the *nuc1-1* context, the number of Nucleolus Organizer Region (NOR) signals increase, and some signals no longer colocalize with the chromocenter structure [38]. In the *nuc1-1* mutant, signals of the AS2 bodies were observed at the inner periphery of the nucleolus: some clusters of AS2 body signals overlapped the chromocenter, whereas some others did not (Figure 4B(a)) [25].

There are still many unclear points about the relationship between the localization of 45S rDNA and those of AS2 bodies. Mutation in the *NUC1* affects the integrity of nucleolar compartments, such as fibrillar centers (FCs), dense fibrillar components (DFC), and granular components (GC), resulting in structural alteration of the nucleolus [38], which may affect patterning of the AS2 bodies. Since the nucleolar subcompartments (GC, DFC, FC) represent coexisting liquid phases, distribution patterns of 45S rDNA and AS2 bodies might be influenced by conditions of liquid-liquid phase separation within the nucleolus of the *nuc1-1* mutant. These results suggested that the localization of AS2 bodies that condense on two peripheral regions of the nucleolus, which partially overlap chromocenters, including 45S rDNA repeats, is important for the function of AS2.

In Arabidopsis, 45S rDNA repeats are localized in tandem arrays at the tops of chromosomes 2 and 4, corresponding to Nucleolus Organizer Regions 2 and 4 (NOR2 and NOR4) [53,54]. Under normal plant growth conditions, the rDNA of NOR4 is transcriptionally active, whereas the rDNA of NOR2 is transcriptionally silenced by repressive chromatin modifications [55,56,57]. We reported that AS2 bodies partially overlap with chromocenters containing 45S rDNA repeats at the periphery of the nucleolus [25].

In the *nuc1-1* mutant, the methylation level of CpG in the upstream of its 28S rDNA (5′ETS region) is reduced as compared with that of CpG in wild-type plants [58]. The CpG methylation levels in the *ETT*/*ARF3* gene are reduced in *nuc1* and *rh10* mutant plants, as well as in the *as2* mutant [20]. The AS2 bodies at the nucleolus periphery might be involved in maintaining the CpG methylation of the *ETT*/*ARF3* gene in developing leaves. It is necessary to clarify whether AS2 is also involved in the maintenance of CpG methylation in genes other than *ETT*/*ARF3*.

We have shown that AS2 bodies partially overlap chromocenters, which contain the 45S rDNA repeats, telomere, and centromere in telocentromeric chromosome [25]. Since AS2 has a zinc finger motif that has the property of binding to DNA, it is necessary to clarify whether AS2 also could bind to CGCCGC sequences in genes other than *ETT*/*ARF3*.

Various bodies have been reported in the nucleus of plants [59,60,61,62,63]. However, to date, there are no reports other than of the AS2 bodies, which are the described bodies that are localized only at two regions of the periphery of the nucleolus. Therefore, AS2 bodies are distinct from other nuclear bodies, and the coordinated roles of these distinct entities require further detailed analyses.

### 3.3. New Roles of the Periphery of the Nucleolus in Plants

The nucleolus has long been known to be the assembly site for ribosomes. In recent years, the nucleolus has been reported to play various roles, such as controlling the cell cycle, responding to various stresses such as viral infection and DNA damage, and detecting the nutritional status of cells [3,8,64,65]. In addition, nuclear lamina and nucleolus peripheral regions are nuclear landmarks that contribute to the inhibition of chromosome assembly, and they are called lamina-associated domains (LADs) and nucleolus-associated domains (NADs), respectively. In human cells, the nucleolus periphery acts as a genomic compartment for repressive chromatin states [27,28]. In Arabidopsis leaves, NOR4 associates with the nucleolus and its rDNA repeats are actively transcribed, whereas NOR2 is inactive and is excluded from the nucleolus [57,66,67]. NADs at the nucleolar periphery in Arabidopsis leaves are detected as condensed chromatins called chromocenters [57,68]. Such condensation is not observed in human cells, suggesting that there is a mechanism for gene expression regulation that is specific to plant cells.

AS2, NUC1, and RH10 are involved in maintaining the DNA methylation of *ETT*/*ARF3*, which might be achieved through at least several independent pathways [14,20]. In addition, AS2 binds specifically the sequence containing CGCCGC in exon 1 of *ETT*/*ARF3*. AS2 localizes to the nucleolar periphery, possibly acting to maintain DNA methylation levels in *ETT*/*ARF3*. In the future, it will be important to identify molecules interacting with AS2 that are necessary for AS2 body formation, and also to understand the mechanisms of epigenetic regulation, like methylation.

*AS2* is expressed at an early stage of leaf primordia [42]. We have shown that AS2-AS1 control the expression of Kip-related protein5 (KRP5) (for inhibitors of cyclin-dependent protein kinases) and Isopentenyltransferase3 (IPT3) (for the biosynthesis of cytokinin) genes through *ETT*/*ARF3* and *ARF4* functions [15,31]. These data suggest that the AS2-AS1-ETT pathway plays a critical role in controlling the cell division cycle and the biosynthesis of cytokinin around SAM to stabilize leaf development in Arabidopsis plants. AS2 bodies are distributed to daughter cells during the progression of the M phase from anaphase to telophase in leaf primordia [25]. In addition, bridge-like structures, which resemble ultra-fine DNA bridges that are not detected via DAPI staining, appear to link the two AS2 bodies [25]. AS2 bodies may function in repressing the *ETT*/*ARF3* gene at perinucleolar domains in cooperation with NUC1, RH10, and RID2 for maintaining the differentiated state of leaf cells (Figure 5C).

Some future tasks are to clarify the mechanism of how the AS2 bodies are inherited, and to understand how NUC1, RH10, and RID2 are involved in the inheritance of the AS2 bodies. The study of AS2 should provide a clue to elucidate the mechanism controlling the maintenance of gene expression for the organ development that is unique to plants.

## 4. Materials and Methods

### 4.1. Plant Materials and Growth Conditions

*Arabidopsis thaliana* ecotype Col-0 (CS1092), and the two mutants *as1-1* (CS3374) and *as2-1* (CS3117), were obtained from the Arabidopsis Biological Resource Center (ABRC). The details of *as2-5* have been described previously [41]. As we described previously [41], we outcrossed *as2-1* with Col-0 three times and used the progeny for our experiments. The details of *snp1* have been described previously [40]. We crossed *as2-1* stamens with *snp1* pistils and analyzed the leaf phenotypes of the first filial generation. The detailed analyses of *rh10-1* [31], *nuc1-1* [52,69], and *rid2-1* [39] were described previously. *rh10-1* and *nuc1-1* were isolated from the Col-0 background, and *rid2-1* was isolated from the L*er* background. For phenotypic analysis, seeds were sown on soil or on agar-solidified Murashige and Skoog (MS) medium. After 2 days at 4 °C in darkness, plants were transferred to a regimen of white light at 50 μmol m^−2^ S^−1^ for 16 h daily at 22 °C, as described previously [41]. The ages of the plants are given in terms of the number of days after sowing (DAS).

### 4.2. RT-PCR

We essentially followed the method that has been described previously [42]. Shoot apices of the mutants and wild-type plants were harvested 14 DAS, which were grown on soil, and then immediately frozen in liquid nitrogen and stored at −80 °C. The total RNA was isolated using the RNeasy Plant mini kit (QIAGEN N. V., Venlo, The Netherlands). Then, first-strand cDNA was synthesized from 2000 ng of the total RNA using a Rever Tra Ace (TOYOBO Co., Ltd., Osaka, Japan) and Oligo (dT) 12-18 primer (Life technologies Corporation, Carlsbad, CA, USA). The real-time PCR was performed in the presence of the Power Track^TM^ SYBR^TM^ Green Master Mix (Applied Biosystems, Waltham, MA, USA). The amplification was monitored in real time using the Step One Plus Real-time PCR system (Life technologies Corporation, Carlsbad, CA, USA). The primers we used for this analysis are listed in Appendix A.

### 4.3. Construction of AS2-YFP, snp1-YFP, as2-5-YFP, as2(Q21T_P22S_E23D)-YFP, and as2(E23D)-YFP

The wild-type AS2-YFP was described previously [23,25]. To amplify the full-length *snp1*, we designed primers: in the forward primer, *Sal*I sites and *Xho*I sites were added upstream of the AS2 start codon, and in the reverse primer, the codons that encode the three glycine and one alanine residues and the *Nco*I site were added as a linker. To amplify the full-length *as2-5*, we designed primers: in the forward primer, a *Xho*I site was added upstream of the AS2 start codon, and in the reverse primer, the codons that code three glycine and one alanine residues and the *Nco*I site were added as a linker. The full-length *snp1* or *as2-5* DNA was amplified via PCR using Pyrobest (Takara Bio Inc., Shiga, Japan) with appropriate pairs of primers, and the fragments were inserted into the *Eco*RV site of pBlueScript SK-.

To construct as2(Q21T_P22S_E23D)-YFP and as2(E23D)-YFP, as2(Q21T_P22S_E23D) and as2(E23D) DNA were synthesized and cloned in the *Eco*RV site of pUC by GenScript Corp. (Piscataway, NJ, USA). The DNA sequences of XhoI-AS2(CmASL3(Q21T_P22S_E23D))-3xGly-NcoI and XhoI-AS2(E23D)-3xGly-NcoI were described in Appendix A.

The plasmids that contain *snp1*, *as2-5*, *as2(Q21T_P22S_E23D),* and *as2(E23D)* sequences were digested by *Xho*I and *Nco*I, and the pEYFP (Clontech, Mountain View, CA, USA) was cut by *Nco*I and *Spe*I. The binary vector pER8 [70] was digested by *Xho*I and *Spe*I. Then, the DNA fragments were separated via agarose gel electrophoresis and isolated from the gel using the Wizard^®^ SV Gel and PCR Clean-Up System (Promega Corp., Madison, WI, USA). EYFP and as2 variant fragments were inserted into pER8 via three-fragment ligation using T4 DNA ligase (Takara Bio Inc., Shiga, Japan).

To introduce the as2 variant-YFPs into *A. thaliana* (Col-0), we used the floral-dip method for Agrobacterium-mediated transformation [71]. T_1_ seeds were grown on agar-solidified (0.8%) plates of MS medium supplemented with carbenicillin (300 μg/mL) and hygromycin (15 μg/mL). Hygromycin-resistant T_1_ seedlings, 14 days after antibiotic selection, were transplanted to soil and cultivated for the production of T_2_ seeds. We established three to five transgenic plant lines that expressed as2 variant-YFP.

### 4.4. Introduction of AS2-YFP into nuc1-1, rh10-1, and rid2-1

We crossed the Col-0 background AS2-YFP stamens with *nuc1-1*, *rh10-1,* and *rid2-1* pistils. The F_1_ seeds were selected on MS medium that contained hygromycin. Then, the F_2_ plants were selected on MS medium supplemented with hygromycin, based on leaf phenotype, via PCR. The appropriate pairs of primers for each mutant that we used for the genotyping are listed in Appendix A. We used the F_3_ plants for the AS2-YFP observation.

### 4.5. Chemicals

4′,6-Diamidino-2-phenylindole (DAPI) and 1,3,5(10)-estratriene-3,17 β-diol (17β-estradiol) were purchased from Sigma (St. Louis, MO, USA). The 17β-estradiol was prepared as a 100 μg/mL stock solution dissolved in dimethyl sulfoxide (DMSO), and the DAPI was a 5.0 mg/mL stock solution dissolved in methanol (Wako, Osaka, Japan). The solutions were stored at −20 °C in darkness. The working solution was made by diluting the stock solution just before use.

### 4.6. Observation of Fluorescence

To examine the subcellular localizations of the YFP fused to AS2 and the as2 variant-YFPs, we established three to five lines of transgenic Arabidopsis plants. To induce the expression of the transgenes, the 6 DAS transgenic Arabidopsis whole plants were soaked in 0.05 μg/mL of 17β-estradiol in 1× MS medium under vacuum conditions for 30 min, and were then incubated at 22 °C overnight. The patterns of fluorescence due to YFP and DAPI were analyzed in 4 to 75 cells of each line of transgenic plants after the induction of expression of transgenes. For DAPI staining, Arabidopsis whole plants were incubated in 4% paraformaldehyde for 30 min under vacuum conditions at room temperature, washed with 1× phosphate-buffered saline (PBS) three times, and stained with 5 µg/mL of DAPI diluted with 1× PBS. Images were recorded via confocal laser scanning fluorescence microscopy with an Objective Plan-Apochromat 63×/1.4 Oil DICII and an Objective α Plan-Apochromat 100×/1.46 Oil DIC (LSM710; Carl Zeiss, AG, Baden-Württemberg, Germany). We acquired z-stack images of the AS2 bodies using Zen 2012 software (Carl Zeiss AG, Baden-Württemberg, Germany).

### 4.7. In Vitro Protein–dsDNA Interaction Assay (AlphaScreen)

We followed the method described previously [20]. Biotinylated oligos (50-mer), non-biotinylated oligos, and the non-biotinylated complementary oligos were obtained from Eurofins Scientific SE (Luxembourg City, Luxembourg). The amplified luminescence proximity homogeneous assay was performed using an AlphaScreen FLAG (M2) Detection Kit provided by Perkin Elmer Inc. (Waltham, MA, USA) [72,73,74] to show the interaction of the AS2/LOB domain of AtAS2 with dsDNA designed from the sequences of the *ETT*/*ARF3* exon 1 (Appendix A). The C-terminal FLAG (DYKDDDDK)-tagged proteins were expressed in wheat germ extract (WGE) from in vitro transcribed mRNA obtained from PCR-generated cDNA [75]. The peptide corresponds to amino acid residues 1–119 of AS2 protein fused with FLAG and hereinafter referred to as AS2D-FLAG. We also expressed a mutated protein in which the cysteine residues C10, C13, C20, and C24 were turned into alanine residues, referred to as as2(4C/4A)D-FLAG, and as2 variant D-FLAG. Relative AlphaScreen signals (Figure 3) were defined as the ratios of luminescence of the biotinylated dsDNA oligos to the non-biotinylated dsDNA oligos. The oligomers we used for this analysis were listed in Appendix A.

### 4.8. In Vitro Pull-Down Analysis (Pull-Down Assay)

We followed the method described previously [20]. We used Dynabeads M-280 Streptavidin-coated magnetic beads (Invitrogen, Waltham, MA, USA) for the in vitro pull-down analysis, which can be used to trap biotinylated molecules for multiple purposes [76,77,78]. We used them to capture biotinylated DNA, and checked whether AS2D-FLAG and as2 variant D-FLAG could bind to this DNA. We used nonbiotinylated oligos to ensure that the DNA did not bind directly to the beads in detectable amounts. The immunoblotting for FLAG-tagged proteins was performed using an anti-FLAG antibody (Wako, Osaka, Japan) and an anti-mouse goat antibody (Cosmo Bio Co. Ltd., Tokyo, Japan). The signal was detected using a SuperSignal™ West Pico PLUS Chemiluminescent Substrate (Thermo Fisher Scientific Inc., Waltham, MA, USA).

## Figures and Tables

**Figure 1 plants-12-03621-f001:**
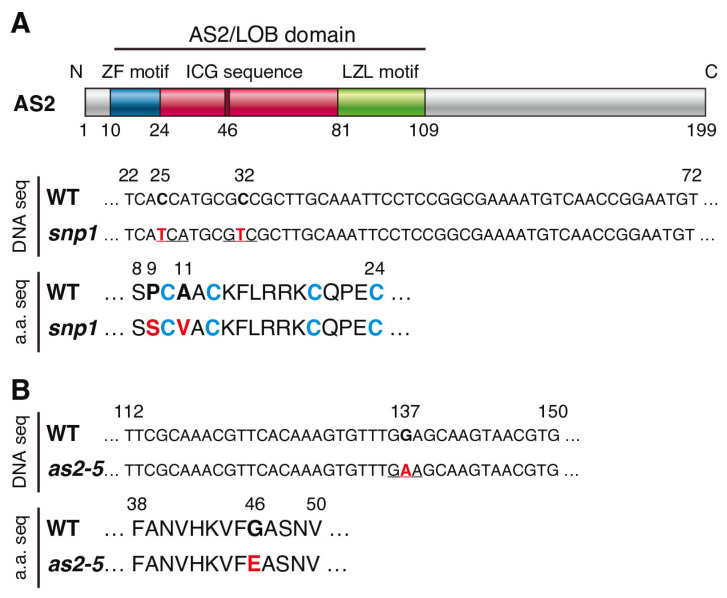
Schematic representation of AS2 protein and amino acid substitutions in *snp1* and *as2-5* mutants. (**A**) Motif and domain organization of AS2, and mutation sites of DNA sequences and amino acid substitutions in the snp1 mutant. (**B**) Mutation site of DNA sequences and amino acid substitution in the as2-5 mutant. The colored boxes for the AS2 protein indicate the AS2/LOB domain: blue depicts the ZF motif, red depicts the ICG sequence, and green depicts the LZL motif. In DNA sequences and amino acid sequences, red letters indicate the base substitutions of nucleotides and the amino acid replacements. Blue letters indicate four cysteine residues in the ZF motif. Each underlining indicates the codon in the DNA sequence that causes the amino acid substitution.

**Figure 2 plants-12-03621-f002:**
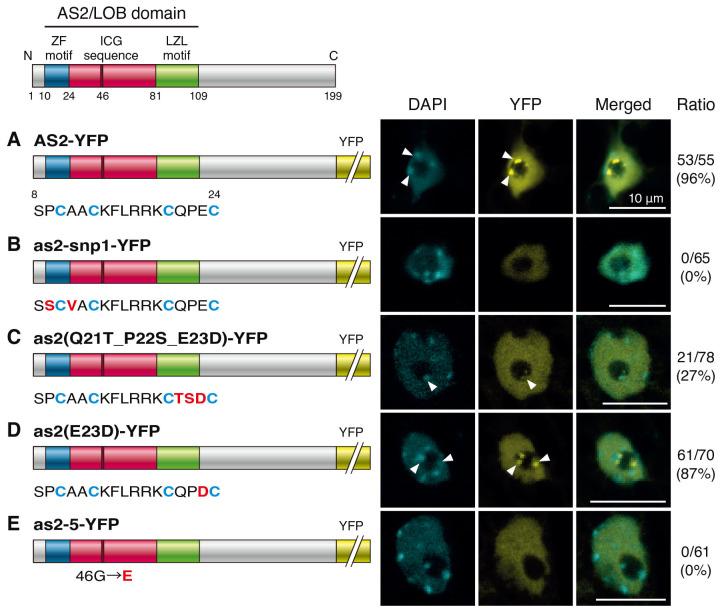
Amino acid residues in the ZF motif and the conserved glycine of the ICG sequence are required for the formation of AS2 bodies. (**A**) Left panel shows schematic representation of AS2-YFP protein with amino acid in the ZF motif. YFP is shown with the yellow box in AS2-YFP. Amino acid sequences in and around the ZF motif of the wild type are shown. Blue letters indicate four cysteine residues in the ZF motif. (**B**) Schematic representation of AS2-snp1-YFP protein; (**C**) as2(Q21T_P22S_E23D); (**D**) as2(E23D) with amino acid residues in the ZF motif. Red letters indicate the amino acid replacements. (**E**) Schematic representation of as2-5-YFP with the 46th glycine residue. Red letters indicate replacement with the glutamic acid residue. In middle panel signals due to DAPI (cyan), those due to YFP (yellow), and merged images in representative cells are shown. The nucleoplasm and the nucleolus were detected as DAPI-positive and DAPI-negative (dark) areas, respectively [23,31]. Numbers on the right side of the images show the ratios of the total numbers of AS2-body-positive cells to the total numbers of YFP-positive cells, which were obtained by adding the numbers of cells from the analysis of each transgenic line. Chromocenters and AS2 bodies that partially overlapped one another are indicated with white arrowheads. Bars, 10 μm.

**Figure 3 plants-12-03621-f003:**
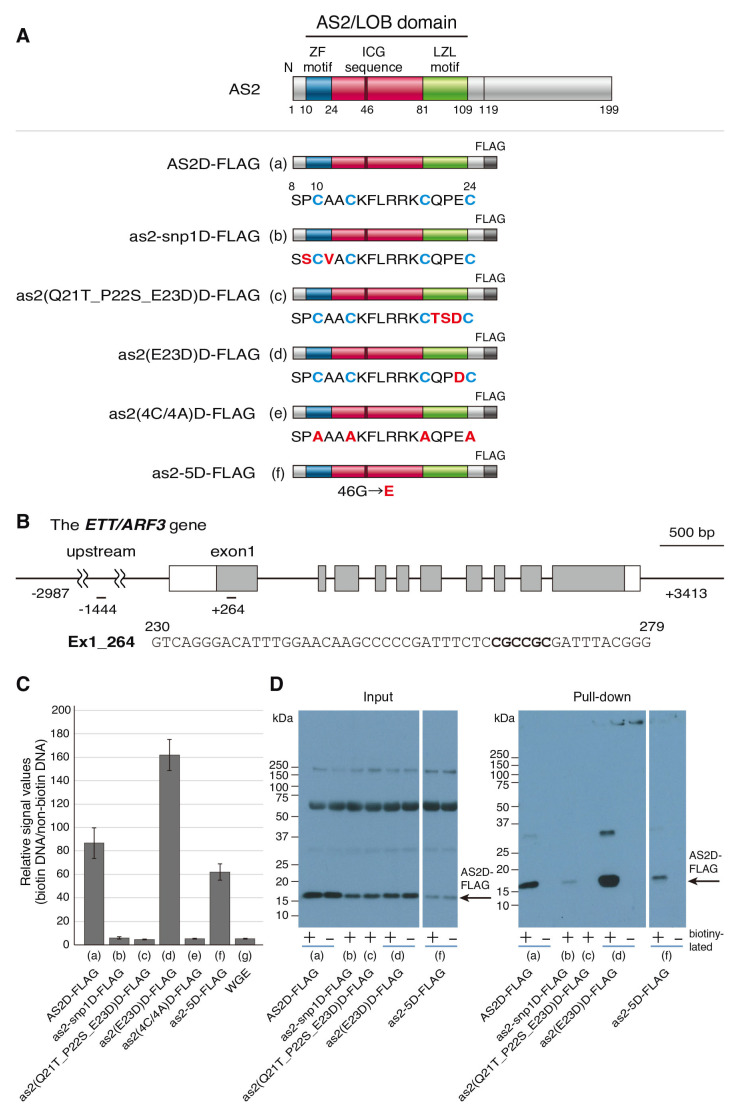
Amino acid residues in the ZF motif are required for interaction with the sequences containing CGCCGC in exon 1 of *ETT*/*ARF3.* (**A**) Motif organization in the AS2 domain (designated AS2D) of AS2 protein. Each as2 variant was fused with a FLAG tag to generate the as2 variant-FLAG. These fusion proteins were synthesized by using the wheat germ protein synthesis system as described in the Materials and Methods. (**B**) Schematic representation of the exon–intron (box-thin line) organization of *ETT*/*ARF3* and the binding sites of AS2 in exon 1. Synthesized DNA (designated Ex1_264) containing genomic sequences were used for the AlphaScreen system. Numbers in the symbols of DNA indicate the genomic distance of the first nucleotides in the binding site from the start codon. The synthesized oligonucleotide was biotinylated at the 5′ end. Non-biotinylated DNA and complementary strand were also synthesized as described in Materials and Methods. (**C**) Relative signal values were calculated as described in Materials and Methods. Bars represent the mean ± SE of triplicate experiments. (**D**) Examination for physical interaction of the as2 variant D-FLAG protein with Ex1_264 DNA. Input lanes show each protein solution before pull-down. Streptavidin-conjugated beads were incubated with biotinylated (+) or non-biotinylated (−) DNA, as indicated below the gel photograph, before incubation with as2 variant D-FLAG proteins. After recovering proteins from streptavidin-conjugated beads, a Western blot with anti-FLAG antibodies was performed to detect the as2 variant D-FLAG protein.

**Figure 4 plants-12-03621-f004:**
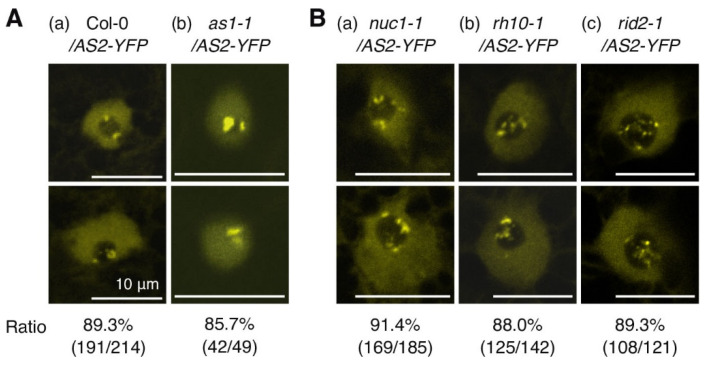
NUC1, RH10, and RID2 are involved in subnucleolar patterning of AS2 bodies. (**A**) Images showing the signals from AS2-YFP fusion proteins in Col-0 and the *as1-1* mutant. (**B**) Images showing the signals from AS2-YFP fusion proteins in *nuc1-1*, *rh10-1*, and *rid2-1* mutants. Expression of AS2-YFP was induced by incubating 6-day-old transgenic Arabidopsis plants (6 days after sowing) with 0.05 μM of 17β-estradiol for 16 h. Signals due to YFP (yellow) are shown; 49 to 245 cells that were YFP-positive were observed in the adaxial epidermis of the cotyledons of each transgenic line, as described in Materials and Methods. Numbers below the fluorescence images show the ratios of the total numbers of AS2-body-positive cells to the total numbers of YFP-positive cells, obtained by adding the numbers of cells from the analysis of each line. Bars, 10 μm.

**Figure 5 plants-12-03621-f005:**
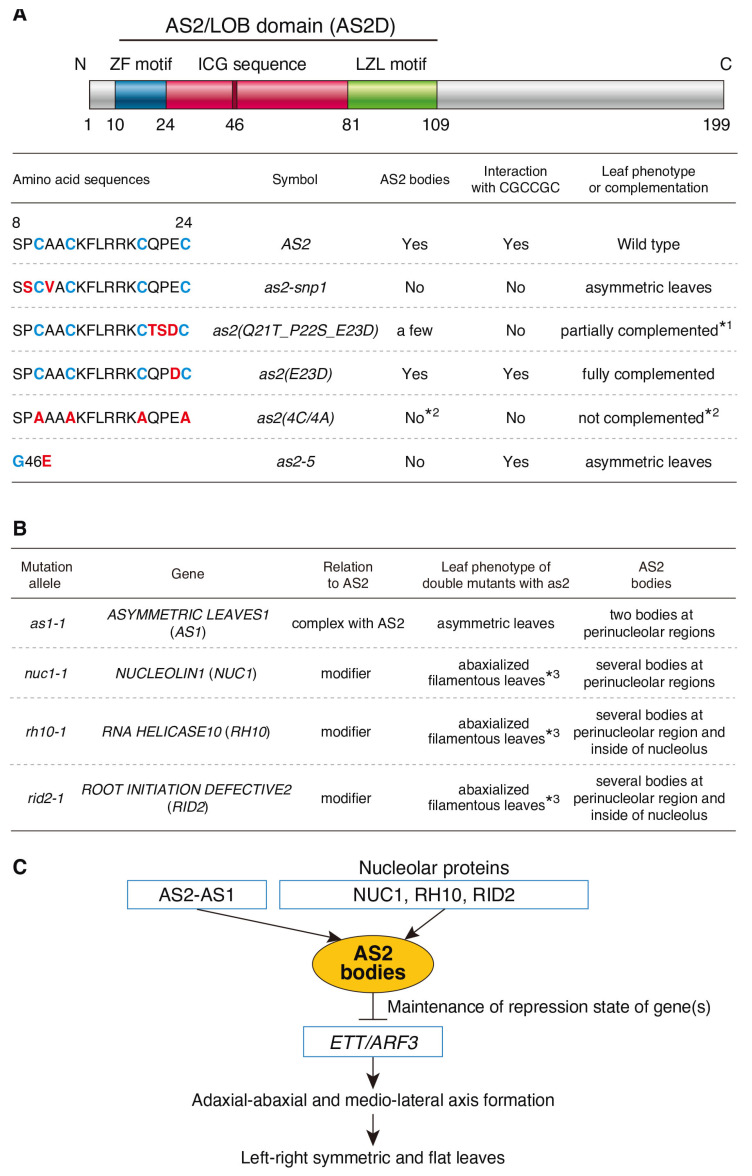
Summary of the functions of amino acid residues in the ZF motif and the conserved glycine of the ICG sequence of AS2, and the roles of NUC1, RH10, and RID2 in the formation of AS2 bodies. (**A**) Summary of the association of mutations in the ZF motif and the conserved glycine of the ICG sequence of the AS2/LOB domain with the ability to form AS2 bodies, and the results of an AlphaScreen experiment with the as2 variants used to examine interactions with CGCCGC sequences. Amino acid residues indicated in red were tested for their importance in the formation of AS2 bodies and the development of normal leaf morphology, such as the establishment of adaxial–abaxial polarity and development of proximal–distal polarity. *1: The length of *as2* leaves in the proximal–distal direction is restored to the wild-type values. However, the phenotype of asymmetric and downwardly curled leaves is not recovered [2]; *2: [25]. (**B**) Summary of AS2 body formation in mutants of genes which affected the leaf phenotype of *as2*. *3: [31]. (**C**) Roles of AS2 and nucleolar proteins (NUC1, RH10, and RID2) in shoot apices during the formation of flat symmetric leaves.

## Data Availability

The data are contained within the manuscript and the Appendix A.

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
