# Peer review of "Arabidopsis ASYMMETRIC LEAVES2 and Nucleolar Factors Are Coordinately Involved in the Perinucleolar Patterning of AS2 Bodies and Leaf Development"

_plants, 2023, doi:10.3390/plants12203621_

Round 1

Reviewer 1 Report

In the manuscript entitled ‘Arabidopsis ASYMMETRIC-LEAVES2 and nucleolar factors are coordinately involved in perinucleolar patterning of AS2 bodies and leaf development’ by Sayuri Ando et al., the authors nicely present that changes in certain amino acid residues close to the Cys residues of the ZF-motif of AS2 (those changes in as2-snp1 and as2(Q21T_P22S_E23D)), affect the formation and the proper localization of AS2 bodies in the perinucleolar region, as well as also affect the interaction of AS2/LOB domain of AS2 protein with the CGCCGC sequence found within exon 1 of the ETTIN gene (an abaxial leaf identity gene repressed by AS2). The study of another as2 mutant (as2-5), which carries a change in the conserved Gly of the ICG- sequence, also leads the authors to conclude that the conserved Gly is required for AS2 bodies formation, but not for interacting with the CGCCGC sequence found within exon 1 of the ETTIN gene. All the above-mentioned changes cause a mutant leaf phenotype, the As2 mutant phenotype, consisting of the development of asymmetric and lobbed leaves. Interestingly, the authors characterize AS2 bodies formation also in mutations of genes required for proper nucleolus formation, and observe an abnormal number of AS2 bodies and their mislocalization. Double mutants involving these mutations and as2 mutations display filamentous leaves. Altogether, these results support the conclusion that proper localization and formation of the two AS2 bodies is linked to proper nucleolar function and morphology, and required for proper leaf development.

Therefore, the authors of this manuscript deepen in the knowledge of the molecular function of AS2 and its three conserved sequences, while providing mechanistic insight regarding the crucial role of certain amino acid residues conserved in AS2 protein for AS2 bodies formation and development of flat symmetrical leaves, and also for the role of such amino acid residues in the interaction of the AS2/LOB domain and the ETTIN DNA sequence.

I consider that this manuscript should be accepted with minor revision.

Suggestions to improve the text:

-ETT/ARF3 is referred to as ETTIN gene only in a couple of places, in the abstract and at the beginning of the introduction, at it may be a good idea to homogenize the way to address at this gene along the manuscript.

-Check and change in the manuscript F1, F2, etc à F1, F2, etc.

-line 66, change ‘the repressive manner of the ETT/ARF3’ to ‘a repressive manner of ETT/ARF3’.

-line 97, referred as à referred to as.

-line 109, formation of AS2 bodies are not yet unknown à formation of AS2 bodies are yet unknown.

-lines 149-150, the same as those à similar to those.

-lines 154, transcript levels of the genes à transcript levels of the previously-mentioned genes.

-line 170, Therefore, this glycine residue à This glycine residue (eliminate Therefore).

-line 215, eliminate mention to Figure 2C.

-line 245, the number of Figure to be mentioned is missing. It should be Figure 3C.

-line 297, include mention to Figure 3C next to AlphaScreen assay.

-line 341, for formation à in the formation.

-line 344, as2 should be in italics.

-line 352, as2-snp1 and as2(4C/4A) refer to alleles and should be in italics.

-line 355, as2(4C/4A) mutant gene à as2(4C/4A) mutant allele

-line 356, these two mutant genes exhibit null phenotypesàthese two mutant alleles lead to null phenotypes (or alternatively, these two mutant alleles cause null phenotypes)

-line 362, AS2 body à AS2 bodies.

-line 400, double mutant with à double mutants with

-line 432, function of the AS2 à function of AS2.

-line 441-442, as well as2 mutant à as well as in the as2 mutant.

-line 476, AS2 should be in italics.

-line 491, maintenance of a gene à maintenance of gene.

-line 535, snp1, as2-5, etc., should be in italics.

-line 535, as2(E23D) à as2(E23D) sequences.

-line 551-552, by these leaf phenotypes à by leaf phenotype.

-line 563, fused à fused to.

-line 567, for overnight à overnight.

-Colors in the images should be changed so that any color-blind person can distinguish the different colors.

-In Figures 2 and 3 it would be very helpful to highlight the 4 Cs in the ZF motif, for example by coloring them in blue, such as in Figure 5.

-Figure 5, wild-type and mutant alleles of AS2 should be in italics.

Reviewer 2 Report

The manuscript continues analysis of the AS2 protein to understand which sequences contribute to localisation as nuclear bodies and binding to the known downstream target sequence in the KANADI promoter. The data presented are informative, clear, and thorough and add to this knowledge of the AS2 and LBD proteins.

Specific comments:

Line 174 lobs should be lobes.

Line 237 Explain “the wild type AS2D protein”, a little more information would be helpful, is it the N terminal 109 amino acids?

Line 268 “whereas when non-biotinylated dsDNAs are used signals might not be detected” the meaning of this statement is not clear.

Line 220: the comparison of AS2 localisation to nuclear bodies is as2 (Q21T P22S E23D) where no localisation is observed versus as2 (E23D) where there is nuclear body

localisation. It is concluded that “It may be required for AS2 body formation that a 23rd amino acid in the third interval is an acidic amino acid.” Maybe I am missing something but the difference between constructs is the change in Q21 and P22. Therefore the conclusion would be better stated according to the importance of Q21 and P22.

Likewise Line 278: “Incubation of the mutant protein as2(Q21T_P22S_E23D) and the dsDNA including 278 Ex1_264 did not generate detectable signal levels(Figure 3Ac, 3Cc), whereas a high signal level was detected in incubation of the mutant protein as2(E23D)D-FLAG and the dsDNA”. The difference in binding is therefore due to changes in amino acid 21 and/or 22. Therefore the conclusion would be better stated according to the importance of Q21 and P22.

Line 320: The data in figure 4 show that there is no change in the ratio of cells with “AS2 body-positive cells to total numbers of YFP-positive cells” but there is a description that the AS2 bodies have changed and the conclusion “These results suggested that NUC1, RH10 and RID2 are involved in the patterning and/or maintenance of AS2 bodies at two domains of the nucleolus periphery.”. It would be helpful to have some quantification of this difference if possible. For instance number of clusters in the nucleus, average size of clusters, and relative location. If there is consistency of the phenotype across all cells in the mutant background, as sample of the nuclei imaged could be further quantified.

Reviewer 3 Report

The manuscript by Ando et al explores some mutations of AS2 important for binding to target DNA and formation of AS2 bodies. They found that QPE motif is required for binding to CGCCGC sequence but can still allow formation of AS2 bodies. Reversely, G46D mutation still enables binding to CGCCGC but totally disrupt AS2 bodies formation. The article is clearly written with many didactic schemes for supporting experiment design, sum up the results and draw hypothesis.

I have marginal comments

- line 171 and 172, it is written “the 46th glycine residue”, which is a bit confusing. The 46th residue is a glycine (the sequence does not have more than 46 glycine). It would be best to write the 46th residue or G46

- line 245, the figure is not numbered (3C)

- would it be possible to predict with Alaphafold what would be the structural difference when QPE is replaced by TSD?
